# Factor Structure of the KABC-II at Ages 5 and 6: Is It Valid in a Clinical Sample?

**DOI:** 10.3390/children9050645

**Published:** 2022-04-30

**Authors:** Gerolf Renner, Dieter Irblich, Anne Schroeder

**Affiliations:** 1Faculty of Special Education, Ludwigsburg University of Education, 71634 Ludwigsburg, Germany; 2Formerly Social Pediatric Center, 55469 Simmern, Germany; d.irblich@t-online.de; 3Werner Otto Institute, 22337 Hamburg, Germany; aschroeder@werner-otto-institut.de

**Keywords:** Kaufman Assessment Battery for Children-Second Edition, KABC-II, confirmatory factor analysis, cognitive assessment, preschool assessment, intelligence test, factorial validity

## Abstract

The factor structure of the German edition of the KABC-II for ages 5 and 6 was examined in a clinical sample. Participants were 450 children ages 5 and 6 who had been assessed due to various behavioral, emotional, or developmental disorders in five Centers for Social Pediatrics (SPCs). Confirmatory factor analyses of the standard test structure including core subtests of the Cattell-Horn-Carroll model and of the Luria model were conducted using maximum likelihood estimation. Several modified structures derived from CHC ability classifications were evaluated. Second-order factor structures corresponding to the standard test structure of the KABC-II demonstrated an adequate global fit for both theoretical models and were superior to unidimensional models. The fit of bifactor models was comparable to second-order models. In all subtests, the general factor accounted for more variance than group factors (broad abilities). However, in more than half of the subtests, unique variance explained the largest portion of the variance. The scale *Learning/Glr* showed a lack of convergent validity. At age 6, a model omitting subtest *Rover* significantly improved the fit. In the combined sample of 5- and 6-year-old children, both second-order and bifactor models with nine subtests demonstrated excellent fit.

## 1. Introduction

The Kaufman Assessment Battery for Children—Second Edition (KABC-II; [1]) is an individually administered comprehensive measure of cognitive abilities for children and adolescents ages 3 to 18 years. The German adaptation [2] is widely used for the assessment of intelligence in clinical settings [3] and special education [4]. The present study focuses on the application of the KABC-II for 5- and 6-year-olds in German Social Pediatric Centers (SPCs). SPCs offer interdisciplinary assessment and intervention for children and youth with developmental disorders, disabilities, chronic illnesses, and psychological problems [5].

The development of the KABC-II was based on two theoretical models: The Cattell–Horn–Carroll (CHC) theory of intelligence [6], and a neuropsychological model inspired by the work of the eminent Soviet neuropsychologist Alexander R. Luria [7].

CHC theory is a psychometrically based descriptive model of the structure of intelligence. Cognitive abilities are organized in a hierarchy of three strata, in its latest version [8] with more than 90 “narrow” abilities (stratum I), 18 “broad” abilities (stratum II), and general intelligence (*g*-factor) as an overarching factor (stratum III). The KABC-II measures five broad abilities: Short-Term Memory (Gsm), Long-Term Storage and Retrieval (Glr), Visual Processing (Gv), Crystallized Ability (Gc), and—for use with examinees ages 7 through 18—Fluid Reasoning (Gf). However, this structure does not consistently follow the CHC theory. Stratum I classifications [1,9] reveal that several subtests are not pure measures of the respective CHC factor (Table 1).

The purpose of the Luria model is to reflect four aspects of mental processing: Sequential Processing, Simultaneous Processing, Learning Ability, and Planning Ability.

The KABC-II consists of 18 subtests. Core subtests are needed to calculate scales and global scales. Supplementary subtests may provide broader coverage of the constructs measured or replace core subtests according to rules provided in the manual. Core subtests are grouped into three to five scales, depending on age and interpretive model (for ages 5 and 6, see Table 1). The names of these scales reflect both theoretical models: *Sequential/Gsm*, *Simultaneous/Gv*, *Learning/Glr*, *Planning/Gf,* and when following the CHC model, *Knowledge/Gc.* The subtest composition of each scale is the same for both models, although they are supposed to measure distinct theoretical constructs. Basically, the Luria model is just a CHC model without *Knowledge/Gc*. Global scores are based on all core subtests and are termed *Fluid-Crystallized Index* (FCI; CHC model) and *Mental Processing Index* (MPI; Luria model), the latter not including *Knowledge/Gc*.

According to the manual [1], the CHC model is the model of choice in most situations because *Knowledge/Gc* is an essential aspect of cognitive functioning. The Luria model should be preferred whenever *Knowledge/Gc* might compromise the validity of the KABC-II as a measure of overall cognitive ability, e.g., when testing children with language disorders.

Factorial validity is an important form of validity evidence [10] and refers to the degree to which empirical data support the putative structure of a test. In multidimensional tests such as the KABC-II, factorial validity is an essential prerequisite for the interpretability of test results. Results of factor analyses should support the relationships between subtests proposed by the theoretical framework. Data on factorial validity show whether a subtest result is primarily determined by the construct suggested by the scale name or by other abilities. If subtests load on multiple factors tests scores cannot be interpreted as measuring a specific construct.

The evaluation of structural models has to be based on the theoretical background of the KABC-II. Specifically, there is a need to know whether scales measure unique and distinct constructs, focusing on a specific target construct, or whether they should be interpreted as a blend of specific and general abilities. At the level of measurement models, the former interpretation is best represented by a bifactor model, the latter by a higher-order model. Bifactor models have gained increasing attention in analyzing the structure of intelligence tests for children, e.g., [11,12]. They allow the various sources of variance to be partitioned between global and specific or group factors. Bifactor models assume that subtest scores are independently associated with the general factor and with group factors [13]. Unlike first-order factors, group factors in bifactor structures are assumed to be uncorrelated with the general factor. Thus, group factors are defined by the shared variance between a subset of indicators once the variance captured by the general factor has been partitioned out.

Unfortunately, the status of scales as described in the manual is somewhat ambiguous. On the one hand, broad abilities are supposed to be “of primary importance for interpreting the child’s cognitive profile” [1] (p. 16) and global scores are considered “almost always secondary in importance to fluctuations within the scale profile” [1] (p. 43). Taking this proposition seriously, we expect test construction to focus on developing unique and distinguishable scales similar to group factors in bifactor models. From a CHC theoretical perspective, incorporating subtests that represent multiple broad abilities should be avoided.

On the other hand, Kaufman and Kaufman [1] did not state that scales and global scales are meant to be uncorrelated, and the scoring rules of the KABC-II imply that scales and global scales share subtest variance. Furthermore, they did not advocate the development of pure measures of broad abilities:

In theory, *Gv* tasks should exclude *Gf* or *Gs*, for example, and tests of other broad abilities, like *Gc* or *Glr*, should measure only that ability and none other. In practice, however, the goal of comprehensive tests of cognitive ability like the KABC-II is to measure problem solving in different contexts and under different conditions, with complexity being necessary to assess high-level functioning. Toward that clinical goal, the authors strove to construct measures that featured a particular ability while incorporating aspects of other abilities [1] (p. 16).

This may appear to be a reasonable approach. Complex academic and real-life challenges that demand intelligent behavior cannot be mastered by applying isolated cognitive functions. Nevertheless, it seems inconsistent to construct subtests that reflect *several* abilities but to interpret scales as indicators of *specific* constructs.

So far, data on the factorial validity have been presented by the test authors and were further investigated in independent studies, mainly reanalyses of standardization data. According to Kaufman and Kaufman [1], the presumed distinction between *Simultaneous/Gv* and *Planning/Gf* could not be substantiated before age 7. Consequently, *Planning/Gf* was not included in the test structure for ages 5 and 6. The final confirmatory factor analysis (CFA) model for core subtests presented in the manual combined ages 5 and 6, including *Rover*, a core subtest for 6-year-olds, but not for 5-year-olds. This model, which does not precisely reflect the standard administration of the KABC-II, demonstrated adequate fit. Still, the average variance extracted (AVE, calculated based on the factor loadings provided in the manual) was low for *Learning/Glr* (0.35) and *Simultaneous/Gv* (0.41). No data on the factorial validity of the Luria interpretive model are provided in the manual.

Reanalyses of the US standardization sample furthered the understanding of the factor structure of the CHC model, though some studies did not consider 5- and 6-year-olds [14,15]. Others focused on different research questions such as the prediction of achievement and did not report specific and detailed results for these ages [16] or included supplementary subtests [16,17,18]. Additionally, the analyses differed in allowing correlated errors or other modifications. Thus, a clear picture concerning the specific factor structure for core subtests at ages 5 and 6 has not evolved from these studies. Various higher-order models aligned with the CHC model showed adequate fit. Potvin et al. [17] combined standardization data from 4- and 5-year-olds and tested an alternative bifactor model, including supplementary subtests. This model fit was good but not superior to the second-order CHC structure. Some results indicate that *Planning/Gf* and *Simultaneous*/*Gv* might be distinct factors at ages 4 to 5 [17] and at ages 6 to 7 [18].

The validation strategy in the German manual of the KABC-II [2] closely followed the procedures described by Kaufman and Kaufman [1]. For four age groups, separate confirmatory factor analyses were conducted, including all subtests (except delayed recall of *Atlantis* and *Rebus*) or core subtests only. German norm groups for ages 5 (*n* = 107) and 6 (*n* = 102) were rather small. Surprisingly, the combined CFA for these age groups is based on only 102 children. *Rover* was included in the analysis, although it is not a core subtest for 5-year-olds and standard scores are not available for this age group. The model presented in the manual showed an adequate fit, but loadings on *Learning/Glr* were small (λ = 0.51), indicating a lack of convergent validity of the respective subtests.

In a reanalysis of the correlation matrices provided in the German manual, Renner [19] found an adequate fit for both the CHC and the Luria model at age 5 but not at age 6. For age 6, an excellent fit could be achieved by omitting *Rover* from *Simultaneous/Gv*. AVE of the CHC model was below 0.50 for *Learning/Glr* and *Simultaneous/Gv* in both age groups, and for *Sequential/Gsm* at age 6.

In a sample of 200 preschool children aged 4 and 5, a CHC-based oblique model and a second-order model were superior to a one-factor solution [20]. The models did not precisely replicate the structure of the KABC-II as both *Face Recognition* (core subtest only for 4-year-olds) and *Pattern Reasoning* (core subtest only for 5-year-olds) were included.

Administering, scoring, and interpreting core subtests is the standard procedure for applying the KABC-II. However, so far, almost all studies have failed to test models that strictly adhere to the test structure at ages 5 and 6. The purpose of the present study was twofold: (1) To further elucidate the internal structure of core subtests for both the CHC and Luria models using confirmatory factor analysis of unidimensional, second-order, and bifactor models; and (2) to extend our knowledge of the KABC-II by providing data on factor structure in a clinical sample of children with heterogeneous developmental disorders. Most studies relied on US standardization data, and no study was conducted in applied clinical settings. However, the Standards for Educational and Psychological Testing [10] require that validity evidence be provided for all intended uses and interpretations of a test. According to Kaufman and Kaufman [1], the KABC-II can contribute to “psychological, clinical, psychoeducational, and neuropsychological evaluations” (p. 8) and informs clinical diagnoses, treatment planning, and placement decisions. These are high-stakes applications. For example, misdiagnoses of intellectual disabilities may have severe long-term consequences for test-takers. Testing children with psychiatric and developmental disorders poses several challenges due to attention deficits, problems in self-regulation, limitations in access skills, test anxiety, etc., which may impact the validity of test results. Test users need to be sure that the proposed interpretation of test scores is equally valid in clinical samples. Nevertheless, the clinical studies presented in the manuals [1,2] did not investigate factorial structure.

Additionally, we wanted to address an issue that has been disregarded in the CFA of common intelligence tests for children and youth. Standard scoring procedures are usually based on equally weighted subtests [16]. The KABC-II standard scores are derived from the sum of scaled scores of subtests, assuming identical contributions of all subtests to scales and global scales. However, in published CFA models of the KABC-II, loadings of subtests on latent factors had not been constrained to be equal. Thus, the meaning of a composite score does not precisely represent the latent variable corresponding to the construct of interest [21,22]. By comparing models with and without equality constraints on subtest loadings, the justification for equally weighting subtests will be considered.

## 2. Materials and Methods

### 2.1. Participants

Data were obtained from clinical records in four SPCs in the southwest (Simmern, Rhineland-Palatinate), north (Hamburg), and northeast (Berlin, Rostock) Germany. Participants were 450 children aged 5 and 6 assessed due to various developmental, behavioral, or emotional disorders. Standards for assessment in SPCs are described in Hollmann et al. [23]. Detailed information on participant characteristics is provided in Table 2. Besides standard scores for subtests and scales of the KABC-II, various demographic variables and medical and psychological diagnoses according to ICD-10 were extracted. Cases were considered only when children had been tested with all core subtests of the KABC-II.

### 2.2. Instrument

Except for some adaptations, the German edition of the KABC-II [2] is closely comparable to the original test. Norms were collected from April 2013 through February 2014. The total standardization sample comprised 1745 children.

In our study, assessments were conducted by experienced clinical psychologists. Test administration and scoring followed the rules given in the German manual [2].

### 2.3. Statistical Analyses and Models

AMOS version 28 [24] was used to conduct CFA with maximum likelihood estimation. Since *Rover* is not included in *Simultaneous/Gv* at age 5, separate analyses were performed for both age groups.

The following models were specified for 5- and 6-year-olds (see also Table 3):Unidimensional measurement models with all core subtests (CHC, Luria) loading on a single-factor (*g*-factor). This model was tested in two variations: Model 1a assumed equal loadings of all subtests on the latent variable. To ensure identifiability, the variance of the latent factor was set to one. Equality constraints were put on all factor loadings. This model corresponds to the standard calculation of global scores (FCI, MPI) by equal weightings of subtests. In Model 1b, equality constraints were released, the standard procedure in confirmatory factor analyses of intelligence tests for children.Second-order (three strata) factor models corresponding to the test structure with one second-order factor and four (CHC) resp. three (Luria) first-order factors. One loading of each factor was fixed to one. Model 2a assumed equal loadings of subtests on first-order factors, and equal loadings of first-order factors on the second-order factor. Model 2b did not contain equality constraints. These models reflect the standard test structure for core subtests of the KABC-II.Bifactor models, with all subtests loading on a general factor and uncorrelated group factors corresponding to the scales of the KABC-II. Model 3a assumed equal loadings of subtests on the general and group factors. Model 3b assumed equal loadings only on group factors, and model 3c did not contain equality constraints. Several latent factors have only two observed variables and are underidentified. To achieve identifiability, the following parameters were fixed to one: (3a) variances of all latent variables, (3b) loadings of one subtest on the general factor and variances of all group factors, (3c) loadings of one subtest on each latent variable, and variances of all group factors.

For age 6, additional second-order and bifactor models were theoretically derived from CHC classifications of narrow abilities represented in the subtests. All these models involve *Rover* and thus were not applicable for 5-year-olds:Both *Riddles* (Gc) and *Rover* (Gv) relate to general sequential reasoning [1,9], a narrow ability subsumed under Gf (Table 1). Model 2c considered this potential association by allowing correlated error terms. In all other aspects, this model was identical to model 2b.In models 2d and 3d, *Rover* was omitted from *Simultaneous/Gv*, as Rover is the only subtest on this scale that has not been assigned to the narrow ability visualization (Table 1).Models 2e and 3e tested whether *Simultaneous/Gv* and *Fluid Reasoning/Gf* could be separated. *Conceptual Thinking* and *Pattern Reasoning* were combined to form Gf, as these subtests are classified as involving the narrow ability induction (Table 1). *Rover* and *Triangles* remained on *Simultaneous/Gv*.

Univariate normality will be assumed for skewness < 2 and kurtosis < 7, as proposed by West et al. [25]. Multivariate normality will be examined by Mardia’s coefficient. SPSS 27 [26] was used for descriptive analyses and one-sample *t*-tests to compare scaled scores with standardization data. As a measure of effect size, Cohen’s *d* was calculated.

To determine the model fit, the χ^2^ test and multiple fit indices were used. These were the comparative fit index (CFI), the root mean square error of approximation (RMSEA), the standardized root mean square residual (SRMR), and the Akaike information criterion (AIC). Higher values indicate a better fit for the CFI and lower values indicate a better fit for the SRMR and RMSEA. Criteria for adequate model fit were CFI ≥ 0.95, SRMR ≤ 0.05, and RMSEA ≤ 0.06 [27,28,29]. Model comparisons were evaluated by χ^2^ difference tests for nested models, AIC (ΔAIC ≥ 4; a threshold indicating “considerably less support” [30] (p. 271)), and Akaike weights [31]. ΔAIC is computed by subtracting the minimal AIC from the AIC for a given model and will be zero for the best-fitting model. Akaike weights are based on AIC and can be interpreted as the conditional probability that a model is the best of several models given the data.

Additionally, average variance extracted (AVE) and coefficient omega (ω; also referred to as construct reliability) were calculated for the best fitting models. Global fit scores can signify an excellent model fit, even when factor loadings are low. AVE allows us to evaluate the convergent validity of subtests of a scale, while ω indicates the extent to which the subtests relate to a given latent variable. AVE ≥ 0.50 and omega ≥ 0.70 will be considered adequate. In bifactor models, the explained common variance (ECV) and omega estimates were computed for the general factor (omega-hierarchical; ω_H_) and the group factors (omega-hierarchical subscale; ω_HS_). For omegaHS, Reise et al. [32] tentatively proposed a minimum value of 0.50. In order to calculate omega estimates, a program provided by Watkins [33] was used. ECV ≥ 0.70 indicates a strong general factor [34].

## 3. Results

### 3.1. Preliminary Analyses

Descriptive statistics of subtests and scales are shown in Table 4, and intercorrelations of subtests are provided in Appendix A. As expected, global scores were significantly lower compared to normative data. One-sample *t*-tests showed medium effects at age 5 (FCI: *t*(249) = −8.91, *p* < 0.001, *d* = −0.63; MPI: *t*(249) = −8.699, *p* < 0.001, *d* = −0.61) and large effects at age 6 (FCI: *t*(249) = −12.65, *p* < 0.001, *d* = −0.89; MPI: *t*(249) = −12.78, *p* < 0.001, *d* = −0.90).

Univariate skewness and kurtosis of subtests fell within the limits proposed by West et al. [25]. Mardia’s coefficient of multivariate kurtosis was 2.88 at age 5 (indicating no departure from normality). A value of 7.93 (critical ratio 3.62) at age 6 signalized that the coefficient of multivariate kurtosis was significantly different from zero. To correct for potential biases of the χ^2^ statistic, the Bollen–Stine bootstrap method [35] was employed for analyses involving 6-year-olds with 2000 bootstrap samples [36].

### 3.2. CHC Models for Age 5

Global fit statistics for all models are displayed in Table 5. For bifactor model 3c, the solution was not admissible due to the negative error variance of *Expressive Vocabulary*. For all other models, non-admissible parameter estimates were not identified.

One-factor models: One-factor models displayed the highest AIC values, and the fit was inadequate according to RMSEA, SRMR, and CFI. The fit of the one-factor model with equality constraints on subtest loadings was worse compared to the unconstrained model (Δχ^2^(8) = 28.19, *p* < 0.001), and both models were significantly inferior to all other models. The loadings of subtests on the general factor are displayed in Appendix A.

Second-order models: Model 2a, assuming equal loadings on the latent variables, was not fully adequate due to RMSEA. Freeing equality constraints led to significant improvement (Δχ^2^(5) = 20.91, *p* < 0.001). Model 2b showed an adequate fit according to all criteria except significant χ^2^.

All standardized loadings of subtests on first-order factors for Model 2b (Figure 1) were statistically significant, with the lowest values for *Atlantis* and *Rebus*. Subtest loadings on the second-order factor ranged from 0.54 (*Atlantis*) to 0.73 (*Riddles*). The second-order factor explained the largest portion of the variance in all subtests. Broad abilities explained 4% to 37% of subtest variance (Figure 2, Appendix A). AVE was adequate for *Knowledge/Gc* and *Sequential/Gsm*, borderline adequate for *Simultaneous/Gv* (0.48), and inadequate for *Learning/Glr* (0.36). The coefficient omega (Table 6) was adequate for all scales except *Learning/Glr* (ω = 0.53). First-order factors showed substantial loadings on the second-order factor (λ ≥ 0.77), and their implied correlations ranged from 0.70 to 0.85 (Table 7).

Bifactor models: For model 3a, assuming equal loadings of subtests on the general factor and the respective group factor, RMSEA failed to meet the criterion. Freeing these equality constraints led to significant improvement (Δχ^2^(8) = 25.50, *p* < 0.001). Model 3b showed an adequate fit according to all criteria (Table 5). All subtest loadings (Figure 1) on *g*, *Simultaneous/Gc*, *Sequential/Gsm*, and *Knowledge/Gc* were significant. Standardized path coefficients of *Rebus* (λ = 0.21, *p* = 0.06) and *Atlantis* (λ = 0.22, *p* = 0.06) on *Learning/Glr* were very low and failed significance. ECV of the general factor was 0.74, while ECV of the group factors ranged from 0.02 (*Learning/Glr*) to 0.13 (*Knowledge/Gc*). The omegaH coefficient for *g* was high (0.83), whereas omegaHS for all group factors, ranging from 0.07 (*Learning/Glr*) to 0.36 (*Knowledge/Gc*), fell below the specified criterion (Table 8).

Model selection: Akaike weights showed that models 2b and 3b had the highest probability of being the best models. Fit indices of these models were almost identical, and ΔAIC was minimal (0.41). Thus, both models can be considered adequate descriptions of the data, though model 3b showed a local misfit with non-significant loadings on *Learning/Glr* and low construct reliability of all group factors.

### 3.3. CHC Models for Age 6

Detailed fit statistics can be found in Table 5. Inadmissible solutions with negative error variances were found for model 3c and the models including *Fluid Reasoning/Gf* (2e, 3e). An admissible solution for 2e could be achieved by constraining loadings on the second-order factor to be equal. For 3e, no more negative error variances were observed after fixing all subtest loadings on *Sequential/Gsm* and *Knowledge/Gc* to one.

One-factor models: Both one-factor models showed significant χ^2^ statistics, did not meet any of the criteria for adequate fit, and displayed the highest AIC values. Again, the fit of the one-factor model without equality constraints on subtest loadings was superior to the constrained model (Δχ^2^(8) = 38.79, *p* < 0.001). Subtest loadings are displayed in Appendix A.

Second-order models: Model 2a could not be retained because of significant χ^2^ statistics and failure to meet the predefined cut-off for RSMEA. Loosening the equality constraints (2b; Figure 3) improved the fit significantly (Δχ^2^(6) = 42.27, *p* < 0.001). The fit of this model could be improved further (Δχ^2^(1) = 4.80, *p* = 0.03; ΔAIC = 2.82) by allowing the error terms of *Riddles* and *Rover* (*2c*) to correlate (*r* = 0.41, *p* = 0.03). Omitting *Rover* from *Simultaneous/Gv* (model 2d) led to an additional improvement of the fit, as indicated by AIC (ΔAIC = 17.71). A five-factor solution (2e) demonstrated inadequate fit.

Subtest loadings on first-order factors for the standard CHC model 2b were statistically significant, with the lowest values for *Atlantis* and *Rebus* (Figure 2). All subtests showed substantial loadings on second-order factors, ranging from 0.67 (*Rebus*) to 0.98 (*Riddles*). The second-order factor explained more variance of all subtests than broad abilities. AVE and omega (Figure 2, Appendix A) were adequate for all scales except *Learning/Glr* (AVE = 0.46; ω = 0.63). Implied correlations between first-order factors range from 0.58 to 0.71 (Table 7). First-order factors load substantially on the second-order factor (λ ≥ 0.73).

For 2d, the best-fitting second-order model, subtest loadings ranged from 0.68 (*Atlantis*) to 0.97 (*Riddles*), AVE was lowest for *Learning*/*Glr* (0.47) and highest for *Knowledge/Gc* (0.77), and omega was above 0.80 except for *Learning/Glr* (0.64). For further data on this model, see Section 3.6.

Bifactor models: Model 3a showed an inadequate fit as indicated by RSMEA and SRMR. Releasing the equality constraints (model 3b; Figure 3) led to a significant improvement (Δχ^2^(10) = 46.41, *p* < 0.001). Further improvement could be achieved with the bifactor configuration (3d) without *Rover*. Model 3e met the criteria for an adequate fit, but AIC was higher compared to 3b (ΔAIC = 17.52) and 3d (ΔAIC = 34.60), and the loadings of *Triangles* on *Simultaneous/Gc* and *Conceptual Thinking* on *Planning/Gf* were insignificant. Additionally, the loadings on group factors changed signs or magnitude depending on the selection of fixed subtest loadings, making the interpretation of results difficult.

Detailed results for 3b are presented in Table 9. All subtest loadings on group factors were significant. Most of the common variance was explained by the general factor (ECV = 0.65). The ECV of group factors ranged from 0.05 (*Learning/Glr*) to 0.12 (*Knowledge/Gc*). The omegaH coefficient for *g* (0.81) surpassed the predefined criterion, whereas omegaHS for all group factors—ranging from 0.19 (*Learning/Glr*) to 0.41 (*Knowledge/Gc*)—did not.

Model selection: Five models (2b, 2c, 2d, 3b, 3d) showed an excellent fit with nonsignificant χ^2^ tests, CFI ≥ 0.996, RMSEA ≤ 0.023, and SRMR ≤ 0.033. Akaike weights favored models 2d and 3d (without *Rover*). Fit indices were almost identical, but in model 3d, the ECV and omegaHS values of group factors were low (ECV ≤ 0.13, ω_HS_ ≤ 0.42).

### 3.4. Luria Models for Age 5

Global fit statistics are shown in Table 10. χ^2^ tests were significant for all models. Inadmissible solutions were not encountered.

One-factor models: Unidimensional models failed to meet the specified criteria for model fit and produced the highest AIC values. The fit of the one-factor model without equality constraints on subtest loadings was marginally superior to the constrained model according to the χ^2^ difference test (Δχ^2^(5) = 11.21, *p* = 0.047). Subtest loadings are displayed in Appendix A.

Second-order models: Model 2a demonstrated an adequate fit according to RMSEA, SRMR, and CFI. Differing from the CHC-based analyses, constraining subtest loadings to be equal did not result in a worse fit. For model 2b, RMSEA was not acceptable. In both models, AVE was satisfactory for *Sequential/Gsm* only, and omega failed to surpass the cut-off value of 0.70 for *Learning/Glr*. Subtest loadings and loadings of the first-order factors on the second-order factor (Appendix A) differed only marginally from the results for the CHC model (Δλ ≤ 0.03).

Bifactor models: Based on RMSEA, SRMR, and CFI, fit was adequate for model 3a. Loadings on group factors (Appendix A) were very low for *Simultaneous/Gv* (λ ≤ 0.26; ω_HS_ = 0.09) and *Learning/Glr* (λ = 0.20; ω_HS_ = 0.06). RMSEA for models 3b and 3c failed to meet the cut-off value.

Model selection: According to Akaike weights, 2a and 3a are the preferred models. Both models fulfill all fit statistical requirements. A χ^2^ difference test (Δχ^2^(2) = 4.38, *p* = 0.112) was not significant, and AIC values were almost identical (ΔAIC = 0.384).

### 3.5. Luria Models for Age 6

Results of the CFA are displayed in Table 10 and Appendix A. The solution for bifactor models 3c was not admissible due to negative error variances. For 3e, an admissible solution was achieved after fixing both loadings on *Sequential/Gsm* to one. Loadings on the general factor are shown in Appendix A.

One-factor models: Model 1a failed to fit the data according to all statistics. Releasing equality constraints significantly improved the fit for model 1b (Δχ^2^(7) = 35.39, *p* < 0.001), but RSMEA, SRMR, and CFI still did not meet the criteria.

Second-order models: χ^2^ tests for all second-order models were nonsignificant, and all models met all criteria for adequate model fit. A χ^2^ difference test, comparing models 2a and 2b, indicated a better fit for the model without equality constraints (Δχ^2^(5) = 20.71, *p* < 0.001). Again, only small differences in subtest loadings and loadings on the second-order factor were found between the Luria and CHC models (Δλ ≤ 0.03). Models 2b and 2d showed an excellent fit. AIC clearly favored model 2d (without *Rover*) over 2b (ΔAIC = 15.56), 2a (ΔAIC = 26.27), and 2e (ΔAIC = 28.50). More details on model 2d are presented in Section 3.6. For model 2b, corresponding to the standard test structure, AVE and omega were adequate for *Simultaneous/Gv* and *Sequential/Gsm*, but not for *Learning/Glr* (Table 6).

Bifactor models: The fit was adequate for model 3a. Global fit statistics were excellent for models 3b and 3d, but negative loadings on *Learning/Glr* and *Sequential/Gsm* were observed. When equality constraints on group factors were imposed by fixing subtest loadings to one, signs changed while all other model parameters and fit statistics remained the same.

Model selection: Akaike weights clearly favored both models without *Rover* (2d, 3d), with a small difference in AIC (ΔAIC = 0.50).

### 3.6. CHC Second-Order Model without Rover

The CHC model for age 6 without *Rover* (2d) had demonstrated excellent fit and corresponded exactly to the test structure of 5-year-olds. Therefore, we decided ad hoc to perform additional analyses, testing the age invariance of this model. After configural invariance was established, measurement weights (subtest loadings) and structural weights (loadings of first-order factors on the second-order factor) were constrained to be equal in both age groups. These constraints did not lead to significant changes in χ^2^ (Table 11).

Finally, we tested this model with the combined sample of 5- and 6-year-olds (Figure 4). The model fit was excellent: χ^2^(23) = 31.70, *p* = 0.107; CFI = 0.995; RSMEA = 0.029 [0.000, 0.052]; SRMR = 0.022; AIC = 75.70. The thresholds for AVE and omega were surpassed for *Knowledge/Gc* (AVE= 0.77, ω = 0.87), *Simultaneous/Gv* (AVE = 0.58, ω = 0.81), and *Sequential/Gsm* (AVE = 0.66, ω = 0.80), but not for *Learning/Glr* (AVE = 0.41, ω = 0.58). As in age-specific analyses, most subtest variance was accounted for by the general factor.

The bifactor model without *Rover* (3b) demonstrated an excellent fit as well: χ^2^(23) = 32.33, *p* = 0.094; CFI = 0.995; RSMEA = 0.030 [0.000, 0.053]; SRMR = 0.022; AIC = 76.33 (Figure 4).

## 4. Discussion

The manual of the KABC-II and several independent studies have reported data on the factorial structure of the KABC-II at ages 5 and 6. These analyses were mostly based on standardization data and have employed various combinations of subtests and age groups, none of which exactly reproduced the standard scoring procedure of KABC-II core subtests. So far, specific validity evidence for the clinical application of the KABC-II has not been provided. However, in Germany—and likely in most countries—intelligence tests are mainly administered in clinical assessment or assessment for special education eligibility. This study intended to provide the first independent evaluation of the KABC-II factor structure for the CHC and Luria models in a mixed clinical sample.

### 4.1. Standard Higher-Order Models of the KABC-II

The structure of the KABC-II is explicated in the manual as a higher-order model with four (CHC model) or three (Luria model) first-order factors. In our data, global fit indices seem to support both models. They were superior to unidimensional structures. For both age groups, at least one CHC and Luria model met the predefined criteria for model fit (Table 5 and Table 10). However, the overall fit of a CFA model does not exclude a local misfit [37] (for an example in the field of intelligence testing, see [38]). A model may demonstrate an excellent fit even if the loadings of indicators on their latent variables are much lower than theoretically expected. Cross-loadings or correlated errors that complicate the interpretation of test scores may remain undetected.

The present results indeed raise some concerns. In both age groups, AVE was low for *Learning/Glr* (≤0.46). *Rebus* and *Atlantis* are supposed to measure the same narrow ability (associative memory) but show a surprising lack of convergent validity. Low correlations between these subtests, notably at age 5, were also reported by Kaufman and Kaufman [1] and in the manual of the German edition [2]. Data for ages 7 to 12 and 13 to 18 in the US and German manuals show that this might be a specific problem in preschool age.

At age 6, modifications derived from CHC classifications show that the fit of the standard CHC model can be improved by allowing error terms of *Rover* and *Riddles* to correlate. Even further improvement could be achieved by omitting *Rover*. This finding is consistent with a reanalysis of the German standardization data [19]. According to the manual, *Rover* was primarily developed to measure executive functioning and “requires a blend of reasoning and visualization” [1] (p. 64). The authors also observed that children used a variety of strategies to solve *Rover*, and even switched strategies during the administration of the test. This multifaceted nature may explain why disregarding *Rover* might contribute to improved model fit. As an adjuvant effect, test users would be less plagued by the many changes in test structure between ages 3 and 6.

For both age groups and both models, the second-order factor accounted for more variance of all subtests than the first-order factor (Figure 2). On average, the general factor explained 41% (age 5) and 40% (age 6) of subtest variance, and broad abilities accounted for 14% (age 5) and 22% (age 6). The interpretive approach outlined in the manual of the KABC-II emphasizes the interpretation of scales. Global scores are supposed to be of “secondary importance” [1] (p. 43). However, our results caution against interpreting KABC-II scales as measures of distinct cognitive dimensions and demonstrate the overall dominance of the *g*-factor. Thus, scores on scales do not exclusively represent the level of competence in a *specific* cognitive skill.

In more than half of the subtests, unique variance explained the largest portion of variance, ranging from 10% to 66% at age 5, and from 5% to 55% at age 6. For *Atlantis* and *Rebus* at ages 5 and 6 and *Triangles* at age 5, more than 50% of the variance was attributable to specificity and measurement error. When utilizing these subtests in isolation (e.g., when incorporated in a cross-battery assessment; [9]), clinicians should be aware that they cannot be interpreted as strong indicators of general intelligence or broad abilities.

McGill [15] noted the absence of information on the structural validity of the Luria model. Loadings of first-order factors on the second-order factor, subtest loadings, AVE, and omega did not markedly differ from the CHC model. Thus, there is no need to worry that the relationships between the remaining first-order dimensions and their loadings on the general factor change when subtests of *Knowledge/Gc* are not included. A modified model, separating *Simultaneous Processing/Gv* and *Planning/Gf*, showed an adequate fit, but was not preferable to the standard model. Nonetheless, as in the CHC model, omitting *Rover* at age 6 led to a substantial improvement in fit.

### 4.2. KABC-II Theory and Bifactor vs. Higher-Order Models

Comparing bifactor and higher-order models of intelligence has instigated controversial scholarly debate. A detailed discussion of the merits and limits of these models is beyond the scope of this paper. Arguments for and against the usefulness of bifactor representations of intelligence have been presented among others by Decker et al. [39], Decker [40], and Dombrowski et al. [41]. Several communalities between bifactor and higher-order models have been mentioned by Brunner et al. [42] and Gignac and Kretzschmar [43].

As pointed out by Eid et al. [44], bifactor models frequently yield anomalous or inadmissible results in empirical applications. In our analyses, negative variances and negative loadings on group factors were observed more often in bifactor than in higher-order models.

Most fit indices did not indicate the superiority of either model. At age 6, the standard Luria bifactor model with *Rover* demonstrated a better fit according to ΔAIC, but for each theoretical model and each age group, the best bifactor and second-order models represented the data equally well. An ideal bifactor structure should be characterized by both a meaningful general factor and meaningful group factors. Whereas the general factor is readily interpreted as *g*, AVE and omegaHS indicate that group factors lack adequate convergent validity. There is not much common variance between subtests once *g* is accounted for. The importance of the *g*-factor was equally obvious in second-order models. These data clearly show that the KABC-II scales cannot be interpreted as unique dimensions independent of *g*. A meaningful interpretation of group factors is hard to conceive and definitely needs additional validity evidence and theory support. As Schneider [21] pointedly wrote: “We care about a sprinter’s ability to run quickly, not residual sprinting speed after accounting for general athleticism” (p. 188).

Discussion of the merits of bifactor and higher-order structures should take into consideration the theoretical background of the KABC-II. As outlined in the introduction, there is a lack of conceptual clarity concerning the status of scales in test development and test interpretation. The manual offers two broad theoretical perspectives, but neither the CHC model nor the Luria model provides a clear rationale for constructing subtests that best represent the respective constructs. From a CHC-theoretical perspective, several inconsistencies are found when looking at stratum I classifications of subtests (Table 1). *Learning/Glr* comprises two subtests that measure associative memory—just one out of more than ten narrow abilities subsumed under Glr, whereas subtests of *Simultaneous/Gv* address five narrow abilities subsumed under two broad abilities. *Rover* was classified as representing three broad abilities [1]. On the one hand, group factors in bifactor models cannot be equated with KABC-II scales. Group factors by definition represent specific constructs that are uncorrelated with general intelligence, whereas KABC-II scales represent a blend of general and more or less specific constructs. On the other hand, interpretation guidelines in the manual focus on specific contents of each scale and deemphasize the influence of the general factor. It is doubtful whether statistical analyses can make sense out of data that lack a clear theoretical background. As noted by Gignac and Kretzschmar [43], the failure to find a distinct dimension in a CFA model may indicate its non-existence, but it may also reflect the insufficient quality of the indicators that define a latent variable.

We propose that clinicians do not base clinical judgment on group factor scores that completely lack content, convergent, and prognostic validity. Nevertheless, recommending not to interpret group factors in bifactor models of the KABC-II may be futile because you cannot interpret them anyway. Standard scores for latent group factors are simply not available, and few practitioners will have the time and expertise to estimate latent scores from observed test scores, as described by Schneider [21].

### 4.3. Separating Planning/Gf and Simultaneous/Gv

At age 6, a five-factorial second-order model separating *Planning/Gf* and *Simultaneous/Gv* produced an inadmissible solution. Though not explicitly stated in the manual, an inadmissible solution for this model was also found in the US standardization sample at ages 7–12, as indicated by a loading of 1.01 of *Planning/Gf* on the general factor. The respective bifactor structure based on core subtests showed adequate fit, but models with four factors were superior.

For ages 7 to 18, similar results have been found by McGill [14]. According to Reynolds et al. [18], Gf and Gv appeared to be distinct constructs for ages 6 to 18. However, in their final models, they allowed cross-loadings of *Pattern Reasoning* (ages 6–18) and *Rover* (ages 6–7) on *Planning/Gf* and *Simultaneous/Gv.* Cross-loadings of *Pattern Reasoning* are also reported by McGill [15]. Distinguishing Gf and Gv has likewise been an issue in several CFAs of the Wechsler Intelligence Scale for Children—Fifth Edition (WISC-V; [45]). Unlike the publishers, several authors [38,46,47] favored a four-factorial solution.

### 4.4. Equally Weighted Subtests

Standard scores for KABC-II scales are derived by equally weighting subtests. Except for the second-order Luria model at age 5, constraining factor loadings to equality showed a significant deterioration of fit. This issue is not discussed in the manual of the KABC-II and, to our knowledge, has not been addressed by previous research on intelligence tests in preschool age. Our data do not allow a closer evaluation of the effects of (non)weighting subtests. Apart from global fit, model parameters were only marginally affected by constraining loadings to equality. Nevertheless, the assumption that equal weights best represent intelligence test data seems questionable at least.

### 4.5. Limitations and Future Directions

The data presented here do not allow generalization to the general population or to other clinical settings. We investigated a highly selected sample. Children could only be included if they had been referred to an SPC—not to other institutions—by a pediatrician or general practitioner and if intelligence testing was deemed necessary by the SPC team. The decision to use the KABC-II was made by the respective examiner, based on the referral question, specifics of the case (e.g., access skills), common institutional practice, and personal preferences. Still, the purpose of our study was not to estimate population parameters but to explore whether the presumed factor structure of the KABC-II and the respective data presented in the manual are compatible with clinical data.

In our study, alternative structures were tested on the basis of CHC theoretical hypotheses only. Other theoretical perspectives might stimulate further modifications. By specifying S-1 bifactor models [44], inadmissible bifactor solutions might have been avoided. In the age groups studied, previous research did not suggest additional empirically based hypotheses. We did not conduct a specification search and may have missed better representations of the data. Yet, relying on modification indices and not on theory is prone to capitalization on chance [48], may result in overfitting [37], and may produce results that are not generalizable [49].

Only core subtests were included in our analyses. This corresponds to standard administration of the KABC-II and clinical practice in the participating SPCs. Consequently, each latent factor corresponding to a KABC-II scale was represented by two indicators only. This is not ideal for conducting confirmatory factor analyses [50] and may lead to unstable parameter estimates and enhances the risk of inadmissible solutions. Three indicators should be regarded as the minimum number to define a latent variable [43,51]. This requirement could not be fulfilled by analyzing KABC-II core subtests except for *Simultaneous/Gv*. In some studies (e.g., [16,17]), supplementary subtests have been included in measurement models of the KABC-II, thus providing three indicators for all first-order factors except for *Planning/Gf*. Regrettably, these models do not correspond to the standard scoring of the KABC-II and therefore are somewhat limited in informing clinical practice where time constraints and attentional and motivational resources of preschool children limit the length of testing sessions.

We propose that future development of multidimensional intelligence tests should be guided by a stronger focus on theory and thorough content analysis of test formats with the purpose of measuring specific constructs.

The importance of factorial validity for test interpretation is obvious. However, factorial validity is essential but not sufficient for the responsible use of multidimensional tests. Few studies (e.g., [16,52]) have addressed other crucial aspects of validity, reliability, and test fairness of KABC-II global scales, scales, and subscales. Specifically, the validity and clinical usefulness of complex interpretive strategies as proposed in the manual of the KABC-II need further evaluation in clinical samples. We suggest that future research place more emphasis on these issues.

## 5. Conclusions

The goal of this study was to present the first independent evaluation of the factorial validity of the KABC-II at ages 5 and 6 in a clinical sample. The results for both the CHC and Luria models are partially consistent with the presumed test structure presented in the manual. Nevertheless, some concerns should be considered in the clinical application of the KABC-II:We recommend that KABC-II scales should not be interpreted as pure measures of specific constructs unrelated to *g*. Although the manual lacks some conceptual clarity, the emphasis on the interpretation of scales—as opposed to global scales—may overshadow the influence of the general factor on subtest scores.Some subtests, notably *Atlantis, Rebus,* and *Triangles* at age 6, are characterized by a large portion of unique variance (Figure 4). These subtests can hardly be interpreted as strong measures of *g* or the respective broad ability when administered in isolation or as part of a cross-battery assessment.Subtests of *Learning/Glr* were found to lack convergent validity, questioning the interpretation of the respective scale as a unitary construct at ages 5 and 6.Separate analyses of the Luria model do not indicate that the omission of *Knowledge/Gc* changed the relationships between the remaining first-order dimensions and their loadings on the general factor.Omitting *Rover* from the CHC test structure significantly improved the model fit. This may be due to the fact that performance in *Rover* is influenced by several narrow abilities, rendering interpretation of this subtest and allocation to a specific scale difficult.

## Figures and Tables

**Figure 1 children-09-00645-f001:**
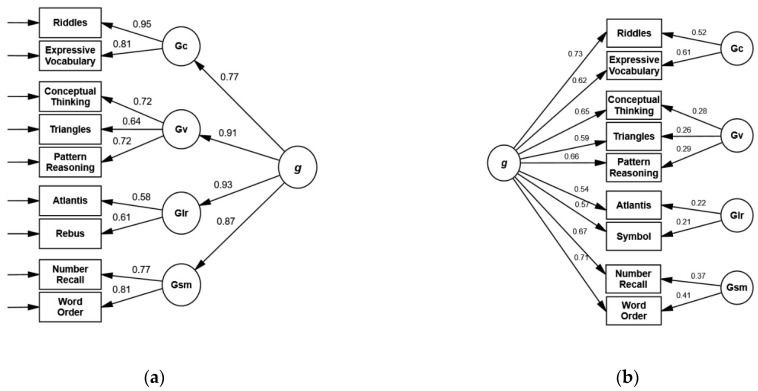
Second-order model 2b (**a**) and bifactor model 3b (**b**) with standardized loading coefficients for the KABC-II CHC subtest configuration at age 5. (**a**) χ^2^ = 38.26, df = 23, *p* = 0.024, CFI = 0.983, RMSEA = 0.052, SRMR = 0.033. (**b**) χ^2^ = 38.64, df = 23, *p* = 0.022, CFI = 0.983, RMSEA = 0.052, SRMR = 0.033.

**Figure 2 children-09-00645-f002:**
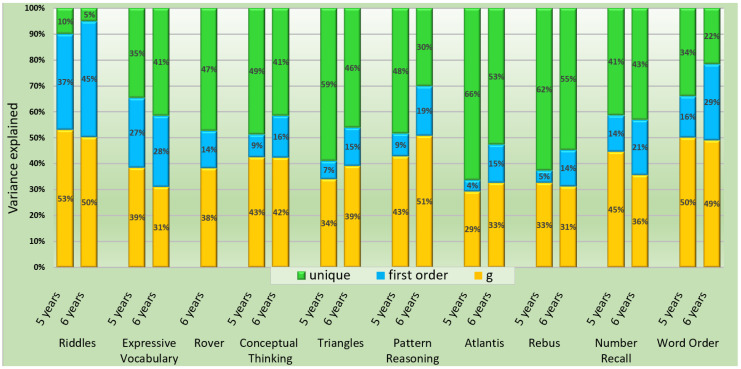
Sources of variance for the KABC-II CHC model core subtest configuration for 5- and 6-year-olds. g = general intelligence.

**Figure 3 children-09-00645-f003:**
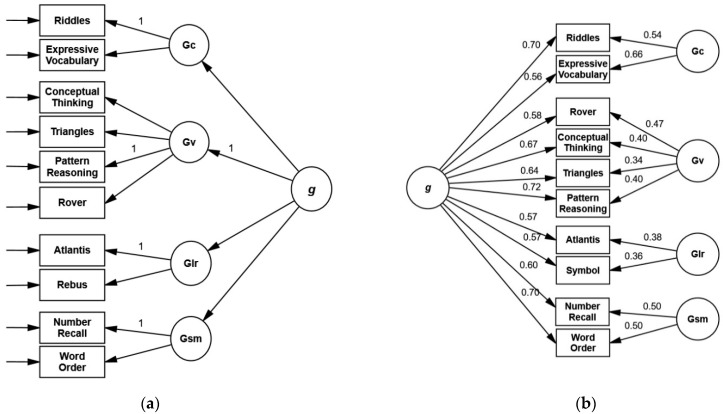
Second-order model 2b (**a**) and bifactor model 3b (**b**) with standardized loading coefficients for the KABC-II CHC subtest configuration at age 6. (**a**) χ^2^ = 34.26, df = 31, *p* = 0.314, CFI = 0.996, RMSEA = 0.023, SRMR = 0.032. (**b**) χ^2^ = 33.55, df = 31, *p* = 0.345, CFI = 0.997, RMSEA = 0.020, SRMR = 0.030.

**Figure 4 children-09-00645-f004:**
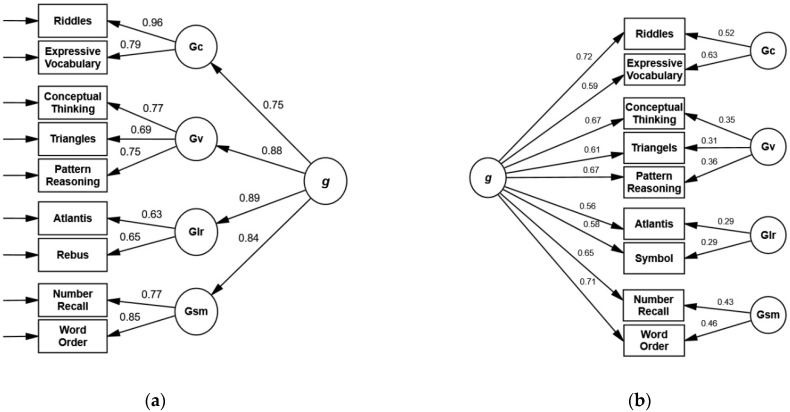
Second-order model (**a**) and bifactor model (**b**) with standardized loading coefficients for the KABC-II CHC core subtest without *Rover* in the combined sample of 5- and 6-year-olds. (**a**) χ^2^ = 31.70, df = 23, *p* = 0.107, CFI = 0.995, RMSEA = 0.029, SRMR = 0.022. (**b**) χ^2^ = 32.33, df = 23, *p* = 0.094, CFI = 0.995, RMSEA = 0.030, SRMR = 0.022.

**Table 1 children-09-00645-t001:** KABC-II scales and core subtests for 5- and 6-year-olds.

Scale Subtest	CHC Narrow Abilities Measured
*Sequential Processing/Short-term Memory (Gsm)*	
Number Recall	Gsm: Memory span
Word Order	Gsm: Memory span Gsm: Working memory
*Simultaneous Processing/Visual Processing (Gv)*	
Conceptual Thinking	Gv: Visualization Gf: Induction
Pattern Reasoning	Gv: Visualization Gf: Induction
Rover (6-year-olds only)	Gv: Spatial scanning Gf: General sequential reasoning Gq: Math achievement ^a^
Triangles	Gv: Spatial relations ^a^ Gv: Visualization
*Learning Ability/Long-term Storage & Retrieval (Glr)*	
Atlantis	Glr: Associative memory
Rebus	Glr: Associative memory
*Crystallized Ability (Gc) (CHC model only)*	
Expressive Vocabulary	Gc: Lexical knowledge
Riddles	Gc: Lexical knowledge Gc: Language development ^a^ Gf: General sequential reasoning

Note. According to Kaufman and Kaufman (2004). ^a^ Abilities not considered relevant by Flanagan et al. (2013).

**Table 2 children-09-00645-t002:** Demographic characteristics of participants.

Variable	*n* (%)
Age	
5;0–5;11	250 (55.6)
6;0–6;11	200 (44.4)
Sex	
Male	306 (68.0)
Female	144 (32.0)
Family structure	
Two-parent family	313 (69.6)
Single-parent family	80 (18.8)
Step-family	28 (6.2)
Foster & residential care	25 (5.6)
Other/unknown	4 (0.9)
Migration	
None	329 (73.1)
Parents only	99 (22.0)
Child	17 (3.8)
Other/unknown	5 (1.1)
Most common psychological diagnoses (ICD-10)	
Intellectual disabilities (F7x.x)	27 (6.0)
Specific developmental disorders of speech & language (F80.x)	111 (24.7)
Specific developmental disorder of motor function (F82)	32 (7.1)
Mixed specific developmental disorder (F83)	154 (34.2)
Attention-deficit hyperactivity disorders (F90.x)	72 (16.0)
Conduct disorders (F91.x)	73 (16.2)
Emotional disorders with onset specific to childhood (F93.x)	55 (12.2)
Most common somatic diagnoses (ICD-10) Endocrine, nutritional & metabolic diseases (E00–E99)	24 (5.3)
Diseases of the nervous system (G00–G99)	16 (3.6)
Diseases of the eye (H00–H59)	28 (6.2)
Certain conditions originating in the perinatal period (P00–P96)	49 (10.9)
Congenital malformations, deformations & chromosomal abnormalities (Q00–Q99)	64 (14.2)

**Table 3 children-09-00645-t003:** Overview of KABC-II subtest configurations for CFA models.

Subtest	1a, b	2a, 2b, 2c				2d				2e					Models 3a, 3b, 3c					Model 3d ^a^					Model 3e ^a^					
	g	F_1_	F_2_	F_3_	F_4_	F_1_	F_2_	F_3_	F_4_	F_1_	F_2_	F_3_	F_4_	F_5_	g	F_1_	F_2_	F_3_	F_4_	g	F_1_	F_2_	F_3_	F_4_	g	F_1_	F_2_	F_3_	F_4_	F_5_
NR	■	■				■				■					■	■				■	■				■	■				
WO	■	■				■				■					■	■				■	■				■	■				
ROV ^a^	■		■								■				■		■			■					■		■			
TRI	■		■				■				■				■		■			■		■			■		■			
CT	■		■				■					■			■		■			■		■			■			■		
PR	■		■				■					■			■		■			■		■			■			■		
ATL	■			■				■					■		■		■	■		■			■		■				■	
REB	■			■				■					■		■			■		■			■		■				■	
RID ^b^	■				■				■					■	■				■	■				■	■					■
EV ^b^	■				■				■					■	■				■	■				■	■					■

Note. Models 2a to 2e include a second-order general factor. Loadings on the general factor are constrained to equality in models 1a, 2a, 3a. Subtest loadings on first-order factors and group factors are constrained to equality in models 2a, 3a, 3b. In model 2c, error terms of ROV and RID are allowed to correlate. ■ = subtest loads on the respective factor. NR = Number Recall; WO = Word Order; CT = Conceptual Thinking; PR = Pattern Reasoning; ROV = Rover; TRI = Triangles; ATL = Atlantis; REB = Rebus; EV = Expressive Vocabulary; RID = Riddles. ^a^ Only 6-year-olds. ^b^ Not included in Luria configurations.

**Table 4 children-09-00645-t004:** Means and standard deviations (SD) for KABC-II subtests, scales, and global scales.

Subtest/Scale	5;0–5;11		6;0–6;11	
	Mean	SD	Mean	SD
Number Recall	7.9	3.2	6.5	3.2
Word Order	8.7	2.9	7.5	3.3
Rover	-	-	8.0	2.6
Conceptual Thinking	8.5	2.9	8.0	3.0
Triangles	8.5	3.1	8.1	3.6
Pattern Reasoning	8.6	2.7	8.3	3.1
Atlantis	9.6	3.1	9.1	2.9
Rebus	8.5	3.1	8.0	3.0
Riddles	8.7	3.4	7.9	3.7
Expressive Vocabulary	9.1	2.9	8.7	3.0
Sequential/Gsm	90.2	15.5	82.9	16.8
Simultaneous/Gv	92.0	15.0	85.9	17.9
Learning/Glr	95.4	14.3	92.4	14.4
Knowledge/Gc	92.8	16.7	89.9	18.9
Fluid-Crystallized Index	90.5	16.8	85.0	16.7
Mental Processing Index	90.7	17.0	83.5	18.3

**Table 5 children-09-00645-t005:** Confirmatory factor analysis fit statistics for KABC-II core subtest CHC configuration.

Model	χ^2^	df	*p*	CFI	RMSEA	90% CI RMSEA	SRMR	AIC	w_i_ AIC
Age 5 (*n* = 250)									
1a g-factor, equal loadings	184.01	35	0.000	0.834	0.131	[0.112, 0.150]	0.074	204.01	0.000
1b g-factor	155.82	27	0.000	0.856	0.138	[0.118, 0.160]	0.061	191.82	0.000
2a second-order, equal loadings	59.17	28	0.001	0.965	0.067	[0.043, 0.091]	0.042	93.17	0.002
2b second-order	38.26	23	0.024	0.983	0.052	[0.019, 0.080]	0.033	82.26	0.545
3a bifactor, equal loadings g & group factors	64.14	31	0.000	0.963	0.066	[0.043, 0.088]	0.050	92.14	0.004
3b bifactor, equal loadings group factors	38.64	23	0.022	0.983	0.052	[0.020, 0.080]	0.033	82.64	0.449
3c bifactor	Solution not admissible								
Age 6 (*n* = 200)									
1a g-factor, equal loadings	249.03	44	0.000	0.773	0.153	[0.135, 0.173]	0.096	271.03	0.000
1b g-factor	210.24	35	0.000	0.806	0.159	[0.138, 0.180]	0.078	250.24	0.000
2a second-order, equal loadings	76.53	37	0.000	0.956	0.073	[0.050, 0.096]	0.056	112.53	0.000
2b second-order	34.26	31	0.314	0.996	0.023	[0.000, 0.059]	0.032	82.26	0.000
2c second-order, errors of ROV and RID correlated	29.44	30	0.495	1.000	0.000	[0.000, 0.052]	0.031	79.44	0.000
2d second-order, 2b without ROV	19.73	23	0.658	1.000	0.000	[0.000, 0.048]	0.027	63.73	0.590
2e second-order, five factors with Gf	Solution not admissible								
3a bifactor, equal loadings, g & group factors	79.96	41	0.000	0.957	0.069	[0.046, 0.091]	0.061	107.96	0.000
3b bifactor, equal loadings, group factors	33.55	31	0.345	0.997	0.020	[0.000, 0.058]	0.030	81.55	0.000
3c bifactor	Solution not admissible								
3d bifactor, 3b without ROV	20.47	23	0.614	1.000	0.000	[0.000, 0.051]	0.027	64.47	0.409
3e bifactor, five factors with Gf	49.10	30	0.015	0.979	0.057	[0.025, 0.084]	0.050	99.07	0.000

Note. CFI = comparative fit index. RMSEA = root mean square error of approximation. CI = confidence interval. SRMR = standardized root mean square residual. AIC = Akaike information criterion. w_i_ AIC = Akaike weights. ROV = Rover. RID = Riddles. *p*-values for age 6 are based on Bollen–Stine bootstrap.

**Table 6 children-09-00645-t006:** Second-order model of KABC-II core subtests: Coefficient omega (ω) and average variance extracted (AVE).

Factor	CHC		Luria	
	ω	AVE	ω	AVE
5-year-olds				
Knowledge/Gc	0.87	0.78		
Simultaneous/Gv	0.74	0.48	0.73	0.48
Learning/Glr	0.53	0.36	0.53	0.36
Sequential/Gsm	0.77	0.63	0.77	0.63
6-year-olds				
Knowledge/Gc	0.87	0.77		
Simultaneous/Gv	0.85	0.59	0.85	0.59
Learning/Glr	0.63	0.46	0.64	0.47
Sequential/Gsm	0.81	0.68	0.81	0.68

Note. Data are based on model 2b.

**Table 7 children-09-00645-t007:** Second-order CHC model: Loadings of first-order factors on the general factor and implied correlations of first-order factors for 5- and 6-year-olds.

Factor	g	Gsm	Glr	Gv	Gc
g		0.87	0.93	0.91	0.77
Sequential/Gsm	0.79		0.81	0.79	0.67
Learning/Glr	0.83	0.66		0.85	0.72
Simultaneous/Gv	0.85	0.67	0.71		0.70
Knowledge/Gc	0.73	0.58	0.61	0.62	

Note. Values are based on model 2b for ages 5 (above the diagonal) and 6 (below the diagonal).

**Table 8 children-09-00645-t008:** Bifactor model of KABC-II core subtests: Factor loadings and sources of variance at age 5.

Subtest	General		Gc		Gv		Glr		Gsm		*h²*
	λ	Var	λ	Var	λ	Var	λ	Var	λ	Var	
Riddles	0.728	0.530	0.518	0.268							0.798
Expressive Vocabulary	0.620	0.384	0.611	0.373							0.758
Conceptual Thinking	0.648	0.420			0.278	0.077					0.497
Triangles	0.592	0.350			0.264	0.070					0.420
Pattern Reasoning	0.660	0.436			0.295	0.087					0.523
Atlantis	0.542	0.294					0.216	0.047			0.340
Rebus	0.573	0.328					0.214	0.046			0.374
Number Recall	0.668	0.446							0.368	0.135	0.582
Word Order	0.709	0.503							0.409	0.167	0.670
ECV	0.744		0.129		0.047		0.019		0.061		
ω/ωS	0.898		0.874		0.734		0.526		0.770		
ωH/ωHS	0.829		0.361		0.119		0.068		0.186		

Note. Gc = Knowledge/Gc; Gv = Simultaneous Processing/Gv; Glr = Learning/Glr; Gsm = Sequential Processing/Gsm; λ = standardized factor loading; *h²* = communality; Var = % variance explained. Values are based on model 3b.

**Table 9 children-09-00645-t009:** Bifactor model of KABC-II core subtests: Factor loadings and sources of variance at age 6.

Subtest	General		Gc		Gv		Glr		Gsm		*h²*
	λ	Var	λ	Var	λ	Var	λ	Var	λ	Var	
Riddles	0.703	0.494	0.539	0.291							0.785
Expressive Vocabulary	0.556	0.309	0.662	0.438							0.747
Rover	0.583	0.340			0.469	0.220					0.560
Conceptual Thinking	0.666	0.444			0.400	0.160					0.604
Triangles	0.637	0.406			0.344	0.118					0.524
Pattern Reasoning	0.722	0.521			0.399	0.159					0.680
Atlantis	0.574	0.329					0.380	0.144			0.474
Rebus	0.569	0.324					0.364	0.132			0.456
Number Recall	0.601	0.361							0.500	0.250	0.611
Word Order	0.700	0.490							0.497	0.247	0.737
ECV	0.650		0.118		0.106		0.045		0.080		
ω/ω_S_	0.922		0.866		0.852		0.635		0.805		
ω_H_/ω_HS_	0.809		0.413		0.236		0.189		0.298		

Note. Gc = Knowledge/Gc; Gv = Simultaneous Processing/Gv; Glr = Learning/Glr; Gsm = Sequential Processing/Gsm; λ = standardized factor loading; *h²* = communality; Var = % variance explained. Values are based on model 3b.

**Table 10 children-09-00645-t010:** Confirmatory factor analysis fit statistics for KABC-II core subtest Luria configuration.

Model	χ^2^	df	*p*	CFI	RMSEA	90% CI RMSEA	SRMR	AIC	wi AIC
Age 5 (*n* = 250)									
1a g-factor, equal loadings	61.39	20	0.000	0.924	0.091	[0.065, 0.117]	0.057	77.139	0.000
1b g-factor	50.18	14	0.000	0.933	0.102	[0.072, 0.133]	0.048	78.181	0.000
2a second-order, equal loadings	27.90	15	0.022	0.976	0.059	[0.022, 0.092]	0.036	53.896	0.518
2b second-order	27.55	11	0.004	0.969	0.078	[0.042, 0.115]	0.036	61.549	0.011
3a bifactor, equal loadings	32.28	17	0.014	0.972	0.060	[0.027, 0.091]	0.043	54.280	0.428
3b bifactor, equal loadings on group factor	27.55	11	0.004	0.969	0.078	[0.042, 0.112]	0.035	61.570	0.000
3c bifactor	23.47	10	0.009	0.975	0.074	[0.035, 0.113]	0.039	59.466	0.043
Age 6 (*n* = 200)									
1a g-factor, equal loadings	127.22	27	0.000	0.843	0.137	[0.113, 0.161]	0.086	145.224	0.000
1b g-factor	91.83	20	0.000	0.888	0.134	[0.107, 0.163]	0.069	123.827	0.000
2a second-order, equal loadings	34.74	22	0.111	0.980	0.054	[0.011, 0.087]	0.045	62.738	0.000
2b second-order	14.03	17	0.764	10.000	0.000	[0.000, 0.053]	0.025	52.029	0.000
2d second-order, 2b without ROV	2.47	11	0.997	10.000	0.000	[0.000, 0.000]	0.011	36.469	0.562
2e second-order, four factors with Gf	26.97	17	0.144	0.984	0.054	[0.000, 0.091]	0.041	64.967	0.000
3a bifactor, equal loadings	36.96	24	0.119	0.980	0.052	[0.009, 0.084]	0.050	60.959	0.000
3b bifactor, equal loadings on group factor	10.89	17	0.862	10.000	0.000	[0.000, 0.035]	0.020	48.894	0.001
3c bifactor	Solution not admissible								
3d bifactor, 3b without ROV	2.97	11	0.994	10.000	0.000	[0.000, 0.000]	0.012	36.969	0.437
3e bifactor, four factors with Gf	Solution not admissible								

Note. CFI = comparative fit index. RMSEA = root mean square error of approximation. CI = confidence interval. SRMR = standardized root mean square residual. AIC = Akaike information criterion. w_i_ AIC = Akaike weights. ROV = Rover.

**Table 11 children-09-00645-t011:** KABC-II core subtests: Invariance of the CHC second-order model without Rover across age groups.

Model	χ^2^	df	Δχ^2^	Δdf	*p*	CFI	ΔCFI	RMSEA	ΔRMSEA	AIC
Configural	57.98	44				0.993		0.024		145.98
Measurement weights	62.77	39	4.79	5	0.44	0.993	0.000	0.023	−0.001	140.77
Structural weights	64.31	36	1.54	3	0.67	0.994	0.001	0.021	−0.002	136.31

Note. CFI = comparative fit index. RMSEA = root mean square error of approximation. CI = confidence interval. SRMR = standardized root mean square residual. AIC = Akaike information criterion.

## Data Availability

Data are available from the first named author on request.

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
