# Peer review of "Factor Structure of the KABC-II at Ages 5 and 6: Is It Valid in a Clinical Sample?"

_children, 2022, doi:10.3390/children9050645_

Round 1

Reviewer 1 Report

The authors are to be congratulated on an excellent review. They have addressed the main authors in the field, and more than 85% of the citations used are from the last 5 years. 
The authors correctly describe the study participants and the sample used, although this should be justified from a literature point of view. In addition, the authors should describe the study population in order to better understand the study sample.
The results you show are fully adequate and relevant to the objective of the study, the pity is not to have been able to appreciate the tables accompanying the discussion of the results.

Reviewer 2 Report

Factor structure of the KABC II at Ages 5 and 6: Is it
Valid in a Clinical Sample?

This paper explores the factor structure of the German edition of the  Kaufman Assessment Battery for Children KABC-II for ages 5 and 6.
A clinical sample was used of 450 children, who had been assessed due to various behavioral, emotional, or developmental disorders in five Centers for Social Pediatrics (SPCs). Confirmatory factor analyses of the core subtests of the Cattell-Horn-Carroll model and of the Luria model were conducted using ML estimation. Second-order factor structures corresponding to the standard test structure of the KABC-II demonstrated adequate global fit for both theoretical models. Fit of two-factor models was comparable to second order models. In all subtests, the general factor accounted for more variance than the scales abilities). However, in more than half of the subtests, unique variance explained the largest portion of variance. The Learning/Glr scale showed lack of convergent validity. At age 6 a model omitting subtest Rover significantly improved fit. In the combined sample of 5- and 6-year-old children, both a second-order and a two-factor model with 9 subtests demonstrated excellent fit.
The work of this paper is mainly statistical, aiming to present the first independent evaluation of the factorial validity of the KABC-II at ages 5 and 6 in a clinical sample. The exploration of all relevant models and the comparisons between are rigorous, and they are performed with the proper focus. The paper, overall, is well written and the whole presentation of the statistical procedure is presented with clarity, so it does not let any question to be asked, regarding the methodological part. The discussion part, the limitation section and the conclusions, elucidate and interpret the findings, while showing the usefulness of this work, which could be characterized as a complete and fine work.
In my opinion, the paper is worth publishing in this form.

Reviewer 3 Report

Thank you for the opportunity to review the manuscript entitled “Factor structure of the KABC II at Ages 5 and 6: Is it Valid in a Clinical Sample?” which was submitted to the journal for peer-review. Even though the manuscript is well-written and provides high levels of transparency and comprehensibility, there are still some minor issues regarding the quality of the present version of the manuscript. Therefore, authors should address the following comments specific to each single section:

Abstract:

  • Lines 12-13: Please rephrase the 3rd sentence. Please note that a theoretical model may at best provide a sound framework for designing instruments rather than including subtests on its own. Then, subtests may reflect some narrow indicators for scales, which are composites or representatives of some kind of broad abilities or latent constructs.
  • Line 18: Please rephrase the sentence as there are no scales within factor analyses. All factor analyses are based on varinave-covariance or correlation matrices from which factors are extracted or prespecified. These factors may or may not be directly associated with those scales included in an instrument. In bifactor models, a general factor as well as specific group factors are associated with their underlying indicators on the same level of inference and independent associations with subtest indicators. 

Introduction:

  • Lines 69 - 72: Please include a short sentence to highlight why factorial validity is an essential prerequisite for the interpretability of test results.
  • Lines 73 - 74: Please rephrase this sentence. A structural (factor) model should always be based on a theoretical framework rather than just consider it.
  • Lines 78 - 79: Please provide a more recent paper pertaining to this matter: 

    Pauls, F. and Daseking, M. (2021). Revisiting the Factor Structure of the German WISC-V for Clinical Interpretability: An Exploratory and Confirmatory Approach on the 10 Primary Subtests. Frontiers in Psychology, 12:710929. doi: 10.3389/fpsyg.2021.710929

  •  

    Page 3, lines 80 - 82: This sentence is confusing and needs to be rephrased. Rather than specifying a higher-order factor as a superordinate general factor (g) that is only associated with and fully mediated by the lower-order factors, bifactor models describe g and the group factors at the same level of inference, featuring independent associations with the subtest indicators. 
  • Pages 3 and throughout the entire manuscript: The use of the term "orthogonal" is not quite unusual when addressing factor characteristics in CFA. Therefore, I would recommend to use "unassociated" or "unrelated" instead.

Materials and methods:

  • Since there is no introduction to the relevance of certain sociademographic or patient-specific characteristics, it is not quite clear what the purpose of Table 2 exactly is. Are ages, sexes (not gender!), and migration distributed as suggested for the clinical subpopulation under investigation? Also, why should information of the family structure be relevant for the scope of the manuscript?
  • Page 6, line 194: The term "norming sample" should be replaced by the term "standardization sample".

Results:

  • Pages 10, 14, and 17: Please include the most relevant fit statistics for both factor models in Figures 1, 3, and 4. This may help to compare both models without having to look at Table 5. Also, standardized paramter coefficients are missing in Figure 3 (a) and should therefore be included.

Discussion:

  • Page 18, lines 513 - 522: Please address the issue of unsufficient variance proportions: what does it exactly mean when only a little variance is accounted for by first-order factors? This has direct implications of whether or to what extent composite scores on those scales representing first-order factors or broad abilties may or may not be interpreted in a meaningful way.  
